# Social Interactions as Recursive MDPs

**Ravi Tejwani\*, Yen-Ling Kuo\*, Tianmin Shu, Boris Katz, Andrei Barbu**
MIT CSAIL & CBMM
{tejwanir, ylkuo, tshu, boris, abarbu}@mit.edu

**Abstract:** While machines and robots must interact with humans, providing them with social skills has been a largely overlooked topic. This is mostly a consequence of the fact that tasks such as navigation, command following, and even game playing are well-defined, while social reasoning still mostly remains a pre-theoretic problem. We demonstrate how social interactions can be effectively incorporated into MDPs (Markov decision processes) by reasoning recursively about the goals of other agents. In essence, our method extends the reward function to include a combination of physical goals (something agents want to accomplish in the configuration space, a traditional MDP) and social goals (something agents want to accomplish relative to the goals of other agents). Our Social MDPs allow specifying reward functions in terms of the estimated reward functions of other agents, modeling interactions such as helping or hindering another agent (by maximizing or minimizing the other agent's reward) while balancing this with the actual physical goals of each agent. Our formulation allows for an arbitrary function of another agent's estimated reward structure and physical goals, enabling more complex behaviors such as politely hindering another agent or aggressively helping them. Extending Social MDPs in the same manner as I-POMDPs (Interactive-partially observed Markov decision processes) extension would enable interactions such as convincing another agent that something is true. To what extent the Social MDPs presented here and their potential Social POMDPs variant account for all possible social interactions is unknown, but having a precise mathematical model to guide questions about social interactions has both practical value (we demonstrate how to make zero-shot social inferences and one could imagine chatbots and robots guided by Social MDPs) and theoretical value by bringing the tools of MDP that have so successfully organized research around navigation to shed light on what social interactions really are given their extreme importance to human well-being and human civilization.

## 1 Introduction

Progress on modeling social interactions and giving machines social goals, such as being particularly nice to a user, is significantly hampered by the lack of theoretical models which characterize what social interactions are. While microsociology has uncovered common structures in social interactions [1] a computational model is still elusive. Until not too long ago, this was also the state of robot navigation and sensing which was revolutionized by extending MDPs [2] to POMDPs [3]. Defining the problem clearly allowed us as a field to understand what we can model and how to do so. Until we take this same step for social interactions, they will remain on shaky ground despite their importance to virtually every interaction humans engage in.

We introduce an extension of MDPs, which we term Social MDPs. In the process, we make several assumptions. First, that agents have both physical goals (e.g., bring the red key home) and social goals (e.g., prevent John from getting his yellow key), and that their overall reward structure is an arbitrary combination of the two, potentially accompanied by other terms. Physical goals are precisely what MDPs can already express, a function of points in a configuration space. Social goals are a function of the estimate of the reward structure of another agent. For example, a reward that hinders another agent is a negative function of the estimated reward of that agent. Complicating matters is the fact that social rewards like beliefs can be recursive: an agent may want to help another agent help them. To model this, Social MDPs are recursive up to a bounded depth, much like interactive POMDPs [4], I-POMDPs. Unlike I-POMDPs, Social MDPs are not recursive in terms of agent's beliefs about the

---

\*Equal contribution.

5th Conference on Robot Learning (CoRL 2021), London, UK.

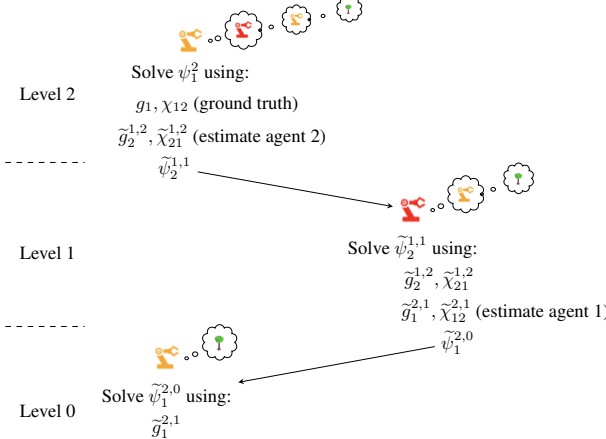

Level 2

Solve $\psi_1^2$ using:

$g_1, \chi_{12}$ (ground truth)

$\widetilde{g}_2^{1,2}, \widetilde{\chi}_{21}^{1,2}$ (estimate agent 2)

$\widetilde{\psi}_2^{1,1}$

Level 1

Solve $\widetilde{\psi}_2^{1,1}$ using:

$\widetilde{g}_2^{1,2}, \widetilde{\chi}_{21}^{1,2}$

$\widetilde{g}_1^{2,1}, \widetilde{\chi}_{12}^{2,1}$ (estimate agent 1)

$\widetilde{\psi}_1^{2,0}$

Level 0

Solve $\widetilde{\psi}_1^{2,0}$ using:

$\widetilde{g}_1^{2,1}$

Figure 1: An example of recursively solving Social MDP for the yellow robot at level 2 in a two-agent interaction scenario. We denote the yellow robot as agent 1 and the red robot as agent 2. At level 2, the yellow robot estimates the red robot's goals (both physical $\widetilde{g}_2^{1,2}$ and social $\widetilde{\chi}_{12}^{2,1}$) and social policy by assuming the red robot is running a level 1 Social MDP. Solving the social policy $\widetilde{\psi}_2^{1,1}$ of the red robot at level 1 requires the red robot to estimate the yellow robot's goals and policy by assuming the yellow robot is running a level 0 Social MDP, i.e., a regular MDP, so we can drop the estimation of $\widetilde{\chi}_{12}^{2,1}$ here. All these estimates are in agent 1's belief space and are updated at every time step.

**Algorithm 1** The algorithm to compute social policy $\psi_i^l$ for agent $i$ at level $l$ and time $t$. We use the estimated social policy $\widetilde{\psi}_j^{i,l-1}$ at previous time step to update the estimated physical and social goal as described in Section 3.3.1. At $t = 0$, we assume $P(\widetilde{g}_j^{i,l,t})$ and $P(\widetilde{\chi}_{jJ}^{i,l,t})$ are from uniform distributions. This algorithm is called at all recursion steps $\widetilde{\psi}_j^{i,l-1}$ to estimate social policy for the other agent $j \in J$. The estimated goals and policies are used to compute the rewards and Q values for selecting the actions.

---

**Require:** $l, s^t, a_J^t, \chi_{ij}, g_i$
  **if** $l = 0$ **then**
    solve MDP for agent $i$
  **else**
    **for all** $\widetilde{\chi}_{ji}^{i,l,t}, \widetilde{g}_j^{i,l,t}$ **do**
      compute
      $P(\widetilde{\chi}_{ji}^{i,l,t}|s^{t-1}, a_J^{t-1})$
      $P(\widetilde{g}_j^{i,l,t}|s^{1:t-1})$
      $\widetilde{\psi}_j^{i,l-1}(s^t, a_J^t, \widetilde{\chi}_{ji}^{i,l,t}, \widetilde{g}_j^{i,l,t})$
    **end for**
    compute $R_i^l(s^t, a_J^t, \chi_{iJ}, g_i)$
    compute $Q_i^l(s^t, a_J^t, \chi_{iJ}, g_i)$
    $\pi_i^l \leftarrow \text{argmax}_{a_i \in \mathcal{A}_i} Q_i^l$
  **end if**

---

state of the world. Instead, Social MDPs are recursive in terms of the rewards of the agents. This makes Social MDPs and I-POMDPs orthogonal and complementary. Social MDPs are specifically formulated to not interfere with the standard extension from MDPs to POMDPs, making it possible to include partial observability. While we do not develop a joint Social I-POMDP here, this is a reasonable extension which would cover far more of the space of social interactions, although one that is computationally challenging.

Our contributions are: (1) formulating Social MDPs where an agent's reward function is an arbitrary function of the recursive estimate of another agent's reward and a physical goal, (2) an implementation where that function is a linear transformation, which captures many notions of helping and hindering, (3) demonstrating that the model performs zero-shot social reasoning in agreement with a human subjects experiment, and (4) examples of the practical utility of recursive social reasoning, In an anonymized online appendix[2] we fully enumerate all possible scenarios predicted by our model given an environment simple enough to allow doing so, demonstrating that it captures a diverse set of social behaviors. We also provide videos of the behavior of our model in all these scenarios.

## 2 Related Work

**Modeling other agents** In order to interact with other agents effectively, an agent must be able to reason about the goals, preferences, and beliefs of other agents [5]. Theory-based models for social goal attribution [6, 7, 8, 9, 10], Bayesian inverse planning to infer an agent's goal given the observations of their behaviors [11, 12], and learning the reward functions of other agents [13] have been explored. Prior research also tried to recognize social interactions such as waving and hugging in videos where people are involved in group activities [14, 15, 16]. These methods generally involve two separate stages [17]: a social perception stage and a coordination or collaboration stage where agents interact. In contrast, Social MDPs constantly reevaluate the goals of other agents enabling them to adapt to changes in the plans of other agents. Social MDPs also allow for enumerating social situations by formally defining the space of what social interactions are, opening the doors to a more theoretical approach to social interactions. Game theory [18, 19] considers altruistic and spiteful behavior through linear combinations of payoffs, similarly to what we consider here although such

---

[2]See https://social-mdp.github.io

models are limited to level 1 reasoning and to scenarios where the goals are known a prior rather than estimated on the fly as they are in Social MDPs and in practical robotic applications.

**Simulating social interactions** Heider and Simmel [20] simulated agent trajectories to study social perceptions through a set of animations involving the movements of geometrical figures. Simulating agent behaviors in physics engines had been explored to collect datasets in fully observable [21, 22] and partially observable environments [23, 17]. The collected datasets are used to study social perception or build machine learning models that can recognize agents' goals. These frameworks assume that each agent has either a physical goal or a social goal. Here we consider a more general scenario where agents have both a physical goal and a social goal and the overall behavior is a blend of the two. This is a more practical setting as when one decides to help another agent one does not abandon every other concern and physical care. Our formulation allows for an arbitrary function of another agent's reward recognizing for complex social interactions. We explore only a subset of the full power of Social MDPs here by considering only linear functions.

**Learning to interact with other agents** Interactive POMDPs [24, 25, 26] (I-POMDPs) are extensions of POMDPs that recursively model the beliefs of other agents. Social MDP and I-POMDPs are orthogonal. Social MDPs allow agents to reason recursively about other agents' reward functions while I-POMDPs allow agents to reason recursively about other agent's beliefs about the state of the world. The two could in principle be combined, but while Social MDPs require solving a modest number of additional nested MDPs, I-POMDPs require significantly nested inference, and when the two are combined the problem quickly becomes intractable. Xie et al. [27] propose a different type of approach that does not require nested inference: learning a low-dimensional representation of another agent's strategy. This approach allows an agent to avoid another agent or to manipulate another agent into some mutually-beneficial behavior. Social MDPs, on the other hand, allow building the strategy of another agent directly into the reward function of an agent, enabling behaviors such as helping or hindering regardless of what the other agent is trying to achieve. Moreover, Social MDPs are zero-shot, while this prior approach is not. From the point of view of generalization and sample-efficient robotics, a zero-shot approach is preferable; in addition, it opens new doors for a more theoretical understanding of social interactions. We could combine Social MDPs with this prior work to build in latent representations of strategies into reward functions [28, 29] creating more efficient approximations of Social MDPs.

## 3 Social MDPs

Social MDPs are recursive MDPs (Markov decision process) with nested estimates of other agent's goals. They are inspired by hierarchical models of games [30] and nested MDP that reason about the beliefs of other agents [31, 32, 33]. Fig. 1 shows an example of recursively estimating the other agent's goals and policy in a two-agent scenario. Like other nested models, e.g I-POMDPs, Social MDPs have the notion of a level. A level 0 Social MDP is simply an MDP: agents reason about the map state. A level 1 Social MDP enables each agent to reason about the physical goals of other agents (those other agents are treated like level 0 agents). A level 2 Social MDP enables each agent to reason about the level 1 social goals of other agents. To perform this nested inference, agents must have access to another agents' physical and social goals. These goals are estimated by solving Social MDPs recursively at every level.

A level 0 agent can take physical actions, but cannot reason socially. A level 1 agent can take actions relative to another agent's physical goals; such actions include helping, hindering, stealing, etc. A level 2 agent can take actions relative to another agent's social and physical goals; such actions include avoiding an attempt to be hindered, recognizing that help is needed, joining in to help together. Levels deeper than 2 continue to describe meaningful interactions although we do not consider them here. It is unclear what level of recursion is required before agents exceed the social reasoning capacities of humans.

### 3.1 Assumptions

As we are in an MDP setting, all physical states are fully observable to all agents. Note however that the goals are not available to other agents. In other words, agents can observe one another at any distance and regardless of any obstacles but they cannot know what another agent wants (either physically or socially) by reading that agent's mind. Agents must infer the physical and social goals, if any, of other agents from their actions. Agents reason completely independently of one another.

In this paper, we consider only pairwise Social MDPs; indeed, all our experiments involve settings where only two agents are present. N-way Social MDPs that consider social interactions between multiple agents at the same time are extensions of pairwise MDP and scale linearly at every level. Computationally however, n-way Social MDPs, as with any such model, scale exponentially in $n$.

As with many multiagent MDP settings, the experiments performed here assume that agents are optimal planners. This is not an inherent limitation of Social MDPs, but relaxing this assumption makes estimating the other agents' goals and rewards far more difficult.

## 3.2 Formal definition of Social MDPs

A Social MDP for agent $i$ with respect to all agents $J$ consists of an arity (here we formulate the pairwise case) and a maximum level, $l$, and is defined as:

$$M_i^l = \langle \mathcal{S}, \mathcal{A}, T, \chi_{iJ}, g_i, R_i^l, \gamma \rangle \tag{1}$$

where $\mathcal{S}$ is a set of states in the environment where $s \in \mathcal{S}$; $\mathcal{A} = \mathcal{A}_J$ is the set of joint moves of all agents in $J$. $a_i$ is an action for agent $i$; $T$ is the probability distribution of going from state $s \in \mathcal{S}$ to next state $s' \in \mathcal{S}$ given actions of all agents in $J$: $T(s' \mid s, a_J)$; $\chi_{iJ}$ is agent $i$'s social goal toward every other agent in $J$. For convenience, $\chi_{iJ}$ is a shorthand for $\bigcup_{j \in J, j \neq i} \chi_{ij}$; $g_i$ is agent $i$'s physical goal; $R_i^l$ is the $l$-th level reward function for agent $i$ based on its estimate of other agents' rewards; and $\gamma$ is a discount factor, $\gamma \in (0, 1)$.

**Reward** Each agent has its own physical goal, e.g., going to a landmark, as well a social goal, e.g., helping or hindering other agents. What enables Social MDPs to go beyond regular MDPs is the recursive nature of the reward function which can be written in terms of the estimated rewards of other agents. The immediate reward of an agent $i$ at level $l$ is computed as follows:

$$R_i^l(s, a_J, \chi_{iJ}, g_i) = r_i(s, a_i, g_i) + \sum_{j \in J, j \neq i} \chi_{ij}(\widetilde{R}_j^{i,l-1}(s, a_J, \widetilde{\chi}_{jJ}^{i,l}, \widetilde{g}_j^{i,l})) - c(a_i) \tag{2}$$

where $r(\cdot)$ is the static reward given the agent's own physical goal $g_i$, $\widetilde{R}_j^{l-1,i}(\cdot)$ is the estimated reward for agent $j$ from agent $i$'s point of view assuming agent $j$ is a level $l-1$ agent, $c(\cdot)$ is the cost for taking an action. For negative levels, the reward is defined to be zero.

$\chi_{ij}$ is the social goal, it transforms the reward of another agent $j$ into a goal that is part of the reward of the target agent $i$. In this paper, we instantiate the model with a linear transformation, so $\chi_{ij}$ is simply a reweighting of the estimated reward of the other agent. If it is a negative value, the target agent will attempt to minimize the reward of another agent, i.e. hindering. A positive value corresponds to helping. Social goals can be eliminated entirely by setting this weight to zero.

In order to estimate another agent's reward function, one needs to estimate that agent's physical and social goals. We use $\widetilde{\chi}_{jJ}^{i,l}$ and $\widetilde{g}_j^{i,l}$ to denote the estimated social and physical goals. The superscript $i, l$ indicates agent $i$ at level $l$ is making the estimations.

We describe how to estimate the social and physical goals in Section 3.3.1.

## 3.3 Planning for Social MDPs

Analogous to MDPs, the Q function of Social MDPs is the sum of immediate reward and the expected value in the future.

$$Q_i^l(s, a_J, \chi_{iJ}, g_i) = R(s, a_i, \chi_{iJ}, g_i) + \gamma \sum_{s' \in S} T(s, a_J, s') V_i^l(s', \chi_{iJ}, g_i) \tag{3}$$

Since agent $i$ is interacting with other agents $j \in J$, it needs to estimate what actions other agents are likely to take in order to compute its state-action value. Social MDPs take the expectation over the estimated goals and actions of agent $j$ to compute $V_i^l(s', \chi_{iJ}, g_i)$:

$$
\begin{aligned}
V_i^l(s', \chi_{iJ}, g_i) &= \max_{a_i' \in \mathcal{A}_i} \left\{ E_{\widetilde{g}_j^{i,l}, \widetilde{\chi}_{jJ}^{i,l}, a_j'} [Q_i^l(s', a_J', \chi_{iJ}, g_i)] \right\} \\
&= \max_{a_i' \in \mathcal{A}_i} \left\{ \sum_{\substack{j \in J, \\ j \neq i}} \sum_{a_j' \in \mathcal{A}_j} \sum_{\widetilde{g}_j^{i,l}} \int_{\widetilde{\chi}_{ji}^{i,l}} \underbrace{P(\widetilde{g}_j^{i,l} | s^{1:t})}_{\substack{\text{estimate physical goal} \\ \text{(Eq. 6)}}} \underbrace{P(\widetilde{\chi}_{ji}^{i,l} \mid s, a_J)}_{\substack{\text{estimate social goal} \\ \text{(Eq. 5)}}} \underbrace{\widetilde{\psi}_j^{i,l-1}(s', a_J', \widetilde{\chi}_{ji}^{i,l}, \widetilde{g}_j^{i,l})}_{\substack{\text{estimate social policy} \\ \text{(Eq. 7)}}} Q_i^l(\cdot) d\widetilde{\chi}_{ji}^{i,l} \right\}
\end{aligned} \tag{4}
$$

When solving agent $i$'s MDP at level $l$, the estimated social and physical goals are further used to update the other agent $j$'s social policy to the actions agent $j$ may take. We denote the estimated social policy for agent $j$ at reasoning level $l-1$ as $\widetilde{\psi}_j^{i,l-1} : \mathcal{S} \times \mathcal{A}_J \times \widetilde{\chi}_{jJ}^{i,l} \times \widetilde{g}_j^{i,l} \to [0,1]$. Algorithm 1 summarizes the steps to compute the state-action values and select optimal actions for any level $l$ at time step $t$. We first update the probability of the estimated goals of other agents using the observed state and the estimated policy from the previous time step. The updated probability of goals are used to update the policy of other agents and compute the reward and Q function of the target agent. The recursion happens at estimating the social policies of other agent at a lower level.

### 3.3.1 Updating social and physical goals of other agents

An agent's estimate of another agent's social and physical goals at time step $t$ and level $l$ can be updated based on the actions performed by the agents. At time step $t = 0$, we use uniform distributions for social and physical goals.

The social goal, estimated at time step $t$, is updated after actions taken by all agents at the previous time step. This update is similar to the belief update in the POMDP framework but based on the estimated social policy of the other agent $j$:

$$P(\widetilde{\chi}_{ji}^{i,l,t} \mid s^{t-1}, a_J^{t-1}) \propto P(\widetilde{\chi}_{ji}^{i,l,t-1} \mid s^{t-2}, a_J^{t-2}) \sum_{\widetilde{g}_j^{i,l,t-1}} P(a_j^{t-1} \mid s^{t-1}, \widetilde{\chi}_{ji}^{i,l,t-1}, \widetilde{g}_j^{i,l,t-1}) \times T(s^{t-1}, a_J^{t-1}, s^t) \tag{5}$$

The physical goal $g_j$ of agent $j$ is estimated by $i$ as follows, similar to Shu et al. [21] but marginalized over the estimated social goal as the agent is estimating the social goal at the same time.

$$P(\widetilde{g}_j^{i,l,t}|s^{1:t-1}) \propto \int_{\widetilde{\chi}_{ji}^{i,l,t}} P(s^{1:t-1}|\widetilde{g}_j^{i,l,t}, \widetilde{\chi}_{ji}^{i,l,t}) \cdot P(\widetilde{g}_j^{i,l,t}) \cdot P(\widetilde{\chi}_{ji}^{i,l,t}) \, d\widetilde{\chi}_{ji}^{i,l,t} \tag{6}$$

### 3.3.2 Estimating social policies of other agents

The $l$-level social policy $\widetilde{\psi}_j^{i,l}$ of the agent $j$ is predicted by $i$ using the Q-function at level $l$-1:

$$\widetilde{\psi}_j^{i,l-1}(s, a_J, \widetilde{\chi}_{jJ}^{i,l}, \widetilde{g}_j^{i,l}) = \text{Softmax}(Q_j^{l-1}(s, a_J, \widetilde{\chi}_{jJ}^{i,l}, \widetilde{g}_j^{i,l})) \tag{7}$$

This is a softmax policy where we use a temperature parameter $\tau$ to control how much the agent $j$ follows the greedy actions. As shown in Eq. 4, in order to use agent $j$'s Q function at level $l$-1, it requires to compute agent $i$'s Q function at level $l$-2, and so on. This involves solving Social MDPs recursively at levels $0, 1, \cdots, l$-1.

### 3.4 Time complexity

The time complexity of solving a Social MDP at level 0 is the same as that of solving an MDP. At level 1, an MDP must be solved for every agent independently in order to compute the likely physical goals of every other agent. Assume that the number of models considered for each pair of agents at each level is bounded by a number $M$ (based on the number of social and physical goals to consider). Solving a Social MDP at level $l$ requires solving $O(M(A-1)^2 l)$ MDPs, where $A$ is the number of agents. Social MDPs form a tree with branching factor $A - 1$ as every agent must compute the pairwise social goal of every other agent until level 0 where the tree bottoms out. There are many potential speedups that can alleviate this runtime to allow for efficient inference even in the face of many agents. For example, a distance horizon could be used where far away agents could simply be considered non-interacting. Similar to Netanyahu et al. [23], it is also possible to speed up the algorithm by amortized inference over goals and relations by training a neural net to recognize goals and relations as initial guesses and refine them through probabilistic inference.

## 4 Results

We apply the Social MDP framework to a multi-agent grid world inspired by previous studies on social perceptions [12, 6, 34]. The $10 \times 10$ world consists of two agents, a yellow robot and red robot, two physical landmarks, a flower and tree, and two objects, a yellow watering can and red watering can. The yellow agent has a low cost for moving the yellow watering can, while it has a high cost for moving the red watering can. Robots can have a physical goal of moving the watering can to a target

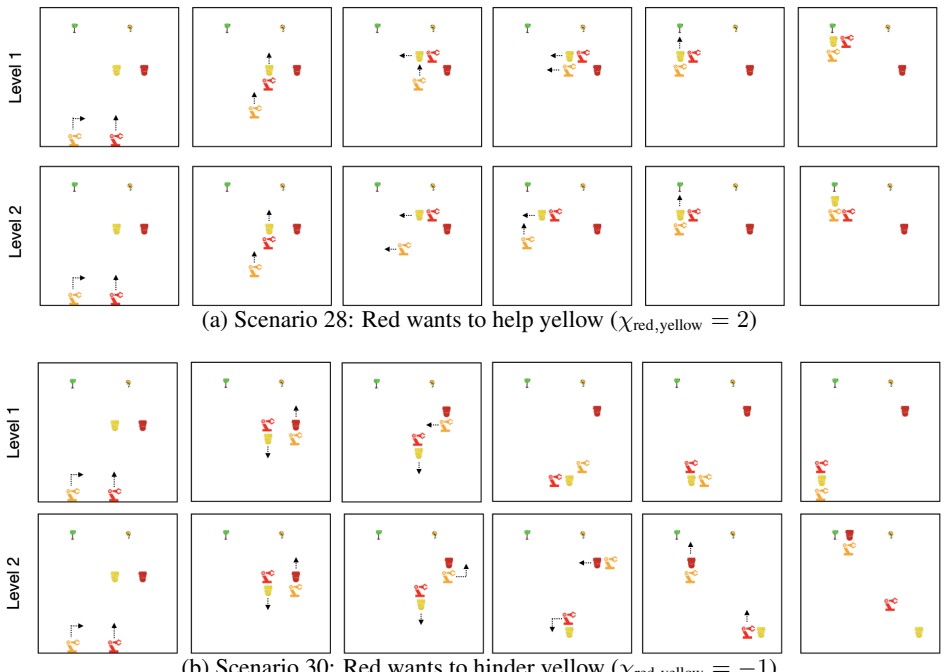

(a) Scenario 28: Red wants to help yellow ($\chi_{\text{red,yellow}} = 2$)

(b) Scenario 30: Red wants to hinder yellow ($\chi_{\text{red,yellow}} = -1$)

Figure 2: Two examples of zero-shot social interactions. The Social MDP gives the robots the ability to understand and predict relationships, thereby making far more efficient actions. The yellow robot wants to water the tree. Moving the yellow watering can is easy for the yellow robot, while moving the red can is hard for the yellow robot. The yellow robot performs inference to understand what the red robot is doing. With a level 1 Social MDP, the yellow robot assumes that the red robot has a physical goal, but not a social goal. With a level 2 Social MDP, the yellow robot assumes that the red robot has both a physical and social goal, then recursively estimates the social goal of the red robot (which is in turn modeled as a level 1 Social MDP).
(a) At level 1, the yellow robot follows the red one around. It does not understand that the red robot is trying to help. The red robot correctly executes its social goal of helping the yellow robot by moving its watering can toward the tree. At level 2, the yellow robot recognizes that red is helping, then estimates where its future trajectory will take it, and efficiently goes to the intercept point accepting red's help.
(b) At level 1, yellow does not infer that red wants to hinder it. As such, it attempts to move the yellow can and repeatedly fails, entering a local minima where the yellow can is the one easiest to move without realizing that the red robot will forever prevent this. At level 2, the yellow robot recognizes that the red robot is attempting to hinder it, gives up on the yellow can, and makes a globally-optimal move of using the harder-to-move red can instead.

plant. Robots can have a social goal of helping or hindering to different degrees. In the grid world, agents can move in four directions (left, right, up, down) or choose not to move.

98 different experiment scenarios [3] are systematically created in this grid world. Each scenario has agents as having either the same physical goal or different physical goals and one of 7 different scaling factors on each of their social goals (-2, -1, -0.5, 0, 0.5, 1, 2) ($2 * 7 * 7 = 98$ scenarios). 2 indicates that the social goal is weighted much more than the physical goal, and an agent wants to maximize the other agent's goal. Similarly for -2, except that an agent wants to minimize another agent's goal. At 1, agents weigh their own social goals equally with their own physical goals. At 0.5, they put twice as much weight on their physical goals as their social goals. These agents are less likely to help, particularly if helping will cost more time or energy than carrying out the physical goal. Finally, either agent can have the factor set to zero; these agents only have a physical goal and no social goal. The fact that we can enumerate all possible social interactions, as predicted by our model, in a given scenario is an important feature missing from alternative representations. All 98 experiment scenarios correspond to reasonable interactions between agents. The degree to which this is true in more complex environments and the degree to which systematically unfolding the model in

---

[3]Interactions for the experimental scenarios can be viewed at `https://social-mdp.github.io/scenarios`

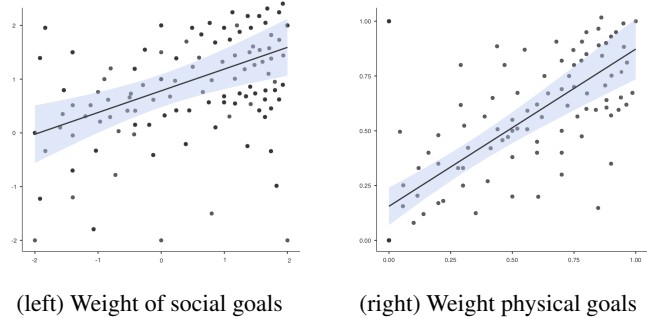

(left) Weight of social goals        (right) Weight physical goals

Figure 3: Twelve human subjects, and our model, the Social MDP, watched and scored 196 videos at different snapshots. These videos consist of 98 scenarios where robots reason at either level 1 or level 2 (presented to the users in randomized order). The straight black line represents the best linear fit to the data, and the light blue band around the line shows the uncertainty in the linear fit. The light blue band represents a 95% confidence interval. (left) Models and humans were asked to predict how social the agents were and the valence of the interaction (was it positive or negative). Non-social settings have a weight of 0, while adversarial settings have a social weight of -2, overwhelming the physical goal of any agent. Humans and machines predict similar social goals both in terms of value and magnitude. (right) Models and humans were asked to predict a weight factor on the physical goal, how much does this agent care about its physical goal. At 0, the physical goal is ignored. At 1, it is weighted equally with a social goal also set at 1. Human and model scores are again highly correlated. Our model is able to effectively generate trajectories that humans recognize as being social interactions. It is also able to predict the type of social interaction that humans believe occurred.

|  | Social MDP (ours) | Inverse Planning | Cue-based |
|---|---|---|---|
| Social Goal | **0.83** | 0.76 | 0.19 |
| Physical Goal | **0.74** | 0.64 | 0.06 |

Table 1: The coefficient of correlation with 95% confidence interval between human and machine judgements for all the 98 experiment scenarios (each scenario has agents having either the same or different physical goals along with one of 7 scaling factors on each of their social goals (-2, -1, -0.5, 0, 0.5, 1, 2)). Refer to Appendix for detailed results for each scenario. We provide two baselines and our own approach. The cue-based model is described in Shu et al. [21]. The inverse planning model is described in Ullman et al. [12]. Social MDPs produce better alignment with ground truth than other models and do not require training like the cue-based model.

more complex environments always results in what humans would describe as social interactions is an important topic for future work.

Each agent's reward for reaching its physical goal is based on that agent's geodesic distance from the goal after taking an action [12]. This physical reward function is parameterized by $\rho$ and $\delta$ that determines the scale and shape of the reward: $r_i(s, a, g_i) = \max\left(\rho\left(1 - \text{distance}(s, a, g_i)/\delta\right), 0\right)$. We set the cost, $c$, of an action $a$, to 1 for grid moves and 0.1 to staying in place while $\rho$ and $\delta$ were set to 1.25 and 5, respectively. The discount factor, $\gamma$, was set to 0.99.

Two of the scenarios selected for our experiment are shown in Fig. 2. We used the Social MDP to select actions for the yellow agent which has a physical goal, watering the flower or tree, while interacting with the red agent. The red agent had a physical goal, watering the tree, and a varying social goal. At every time step, the yellow agent estimates the physical and social goal of the red agent, depending on the Social MDP level. At deeper levels, the agents behave more optimally and more socially.

To quantitatively establish the quality of the social inferences made by the Social MDPs, we compare human judgements of 12 subjects against those of two baseline models: inverse planning [12] and a recent cue-based model [21]. Refer to Table 2 and Table 3 in the appendix for detailed results for each scenario. Humans and models had to estimate the physical and social goals of agents in these environments when the agents were acting both as level one agents (unaware that the other agents are also social) and as level two agents (who could account for the fact that the other agents are social). In Fig. 3 we show the raw judgements of humans and of our models, along with a best linear fit. The performance of all models against human judgements, was measured through correlation coefficient at 95% confidence level, for social goal estimation (r = 0.89 for the Social MDP vs. r = 0.81 for the Inverse Planning model vs. r = 0.23 for the Cue-based model) and physical goal estimation (r = 0.78 for the Social MDP vs. r = 0.72 for the Inverse Planning model vs. r = 0.08 for the Cue-based model).

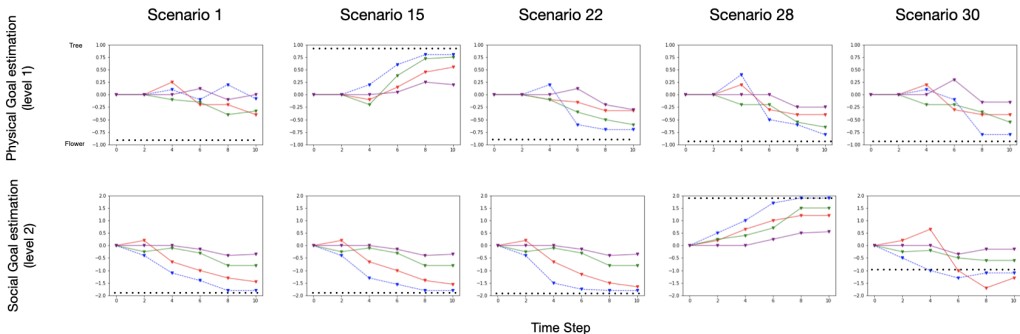

Figure 4: A deep dive into how humans and each model interpret the five experiment scenarios (refer to Appendix for results for all experiment scenarios) at both levels at each time step (in red Human scores, in blue our Social MDP scores, in green Inverse planning [12], in purple the cue-based model [21], and **in dotted the ground truth**). The goal of each model is to interpret how one agent perceives another. (top) At level one, an agent has a belief over the physical goal of another agent. Humans and models predict what this belief is (the degree to which the agent believes that the other agent is heading toward the tree or the flower). Note that all models perform rather well and follow human judgements. (bottom) At level two, an agent has a belief over the physical and social goals of another agent. Humans and models predict what the beliefs of the agents are about the social goals of other agents. In other words, to what degree does this agent think that the other agent is hindering or helping them. Here our model fits human data much better because of its recursive nature. At deeper levels, our model is capable of capturing social interactions and social inferences that other models cannot. Other models are confused, and so predict that there is a very weak or non-existent social goal in most cases while our model follows human judgements.

Our model performs considerably better than other models. This is even more evident in the deep dive shown in Fig. 4. For level one agents, agents that are social but that assume that other agents are not social, all models agreed with human judgements. Yet, for level two agents, agents that are social and can assume that other agents are also social, our models are far better aligned with human judgements. Prior work could capture a far smaller space of social interactions than the Social MDPs we craft here.

# 5    Conclusion

Social MDPs are a first step toward a theory of social interactions that fits within the established frameworks we have in robotics. They can perform zero-shot social recognition and planning for diverse situations. The fact that MDPs can be extended in a natural way that is also computationally tractable to account for many social interactions by nesting inference and allowing models to take arbitrary functions of the estimated rewards of other agents has not been noted before. Our experiments clearly show that Social MDPs are superior to prior models and account for more social interactions.

In this work, we have only begun to explore what Social MDPs can represent. The environment we consider is very simple, yet, at the same time, more than enough to differentiate Social MDPs from other models. So far, we have unrolled Social MDPs only two levels; what exists at deeper levels is still unclear. It is likely that humans do not perform deeply-nested recursive reasoning to carry out social interactions, although, what the cutoff is, and if Social MDPs are close enough to a human's mental model to allow for measuring that cutoff is unknown.

We would like to see in the future that any MDP-based system can be augmented to be social by a straightforward extension with Social MDPs. Much like virtually any approach can be easily augmented to partially-observed environments using POMDPs. Social MDPs and POMDPs are compatible, exploring their combinations and the implications of partial observability for social interactions remains as future work.

**Acknowledgments**

This work was supported by the Center for Brains, Minds and Machines, NSF STC award 1231216, the MIT CSAIL Systems that Learn Initiative, the CBMM-Siemens Graduate Fellowship, the MIT-IBM Watson AI Lab, the DARPA Artificial Social Intelligence for Successful Teams (ASIST) program, the United States Air Force Research Laboratory and United States Air Force Artificial Intelligence Accelerator under Cooperative Agreement Number FA8750-19-2-1000, and the Office of Naval Research under Award Number N00014-20-1-2589 and Award Number N00014- 20-1-2643. The views and conclusions contained in this document are those of the authors and should not be interpreted as representing the official policies, either expressed or implied, of the U.S. Government. The U.S. Government is authorized to reproduce and distribute reprints for Government purposes notwithstanding any copyright notation herein. We would like to thank the members of InfoLab, particularly Bennett Stankovits and Dylan Sleeper, for helping in data collection and environment setup.

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
