# OpenReview forum: "Social Interactions as Recursive MDPs"
_robot-learning.org/CoRL/2021/Conference — CoRL2021 Poster_

### Official Review · Reviewer_jXz2 · 2021-07-20

**Originality:** Very Good
**Technical Quality:** Fair
**Clarity Of Presentation:** Good
**Impact:** 2

**Recommendation:**

Weak Accept: I recommend accepting the paper, but will not argue for my recommendation if the majority of other reviewers have a different opinion.

**Summary:**

This paper presents a novel methodology to include social goals in a decision-making problem (where physical goals are already considered). In this way, the problem is very well defined: how to incorporate social interactions into MDPs? The authors propose to consider recursive goals (social and physical) of all agents of the environment. In doing so, the authors (probably) opened a new research line in the Robotics community: Social Markov Decision Processes (Social MDPs).


**Issues:**

Abstract
The authors consider hindering and helping as social interactions, while one can state that these are more collaborative/uncooperative interactions. Could you define social interactions? For instance, in the experiments you show results with two robots. Do two robots comprise a social group? Perhaps, reasoning about other's intention may be considered as a social behavior, but it is an instance of social behaviors.

In the Abstract you should define the abbreviations: MDP - Markov decision process, and POMDP- Partially observable Markov decision process. The first time MDP is defined is in Sec. 3, line 103.

Section 1. Introduction
You could define better (crisper) physical and social goals by including practical examples.

The authors consider social and physical goals at the same level of abstraction. However, one may figure cases where social goals drive physical goals. In these cases, social goals trigger physical goals. We can even consider, in those specific situations, that the physical goals are drives to satiate the social goals. Throughout the author's description, one can see that both concepts are disentangled, which is not (always) true - e.g. An agent wants to go towards a human to greet him (social goal); so, the agent needs to go to a certain location (physical goal) in order to greet the human. Hence, the physical goal stems from a social goal. How do you see that these cases affect your formulation? For instance, should the physical goal depend on the social goal? If so, how?

Section 2. Related Work, Simulating social interactions
Rewrite lines 82-83 (word repetition - "allow").

Section 3. Social MDPs
Line 43: "Unlike I-POMDPs, Social MDPs are not recursive in terms of agent's beliefs (...)". Line 104: "Social MDPs are (...) and nested MDPs that reason about the beliefs of other agents". Please be coherent or clarify better.

Regarding the notion of levels, in which extent levels deeper than 2 vary from the considered ones?

Section 3.1 Assumptions
IRL should be defined and discussed. There was no previous context when "IRL" was mentioned.

Section 3.2 Formal definition of Social MDPs
Why the physical goal is not part of the definition? Or seen from a different perspective why the social goal is part of the definition? Being coherent here is important.

Reward
In the formula, the expresion R_j^{j-1,i}  is not clear, do you mean R_j^{l-1,i}? It is the estimated reward for agent from 's point of view and following the methodology's formulation, j's are all considered with one level below the target's one.

Sections 3.3. ,3.4, and 3.5
Mathematical choices and formulas should be explained, discussed or referenced.

4. Results
The results are extremely important and show how fruitful was the presented work. All images and plots must be bigger, otherwise one cannot see the real impact of your approach (the videos are also very good, but try to increase the size of the images, perhaps the inclusion of an Appendix is a great solution). In the caption of figure 1. (b) there is a mistake: the red robot wants to hinder the yellow, not to help it. Again, one can envisage social behaviors included in this system, but for now it is not clear that two robots helping/hindering each other is the representation of a social interaction.

Finally, you reference Figure 3, but it is a Table. Please revise it.

Section References
References 12, 19, and 30 do not contain the respective conference or journal, Please include this information.

**Reviewer Expertise:**

Good: General knowledge of the area

**Strengths And Weaknesses:**

* Strengths
Indeed, the novelty of this work is its strength. There are few related works, so the authors may open here a new research line in Robotics. In addition, the problem has paramount importance for topics such as Human-Robot Interaction and Robot Learning.

* Weaknesses
However, the authors define "Social Interactions" in a very superficial way (helping/hindering), which should be comprehensively exploited. Other weaknesses that can be pointed out are:
     - mathematical formulations poorly justified/explained
     - results visually unclear but that are very important to understand the evidences/hypotheses presented in the document

See also "Issues"

**Summary Of Recommendation:**

Although the article is well written and the idea is very interesting, there are some missing links found throughout the document. The document needs some clarifications and discussions (maybe proofs, if that's the case) in the mathematical formulations. There is space for an Appendix for more explanations or additional material.

---

> ### Author Response · Authors · 2021-08-30
> **Response to Reviewer jXz2 (Part 1 of 3)**
>
> Thank you for the positive feedback. We have addressed all of the concerns brought up in the review below. In addition to the detailed response below we have also updated the manuscript fixing the typos the reviewer pointed out.
>
> > Abstract: The authors consider hindering and helping as social interactions, while one can state that these are more collaborative/uncooperative interactions. Could you define social interactions? For instance, in the experiments you show results with two robots. Do two robots comprise a social group? Perhaps, reasoning about other's intention may be considered as a social behavior, but it is an instance of social behaviors.
>
> The reviewer’s point is well taken. There are no satisfactory mathematical definitions of what a social interaction is. That is precisely the problem we aim to address, taking a first step in that direction. We choose hindering and helping as social interactions by following 7 decades of work in the field of micro social interactions (see Maiwald & Suerig, 2017, Microsociology: A Tool Kit for Interaction Analysis for an extensive review). In the field of microsocial interactions studies do consider interactions between dyads (pairs of agents), including hindering and helping as we do here, but without a computational model.
> We believe that a theory-driven approach which begins with work done in the social sciences can both result in robots that are socially adept and eventually elucidate the muddled picture of social interactions that the reviewer points out.
> We have added a more explicit link to microsociology and the long history of these questions (Refer line 31 at page 1)
>
> > In the Abstract you should define the abbreviations: MDP - Markov decision process, and POMDP- Partially observable Markov decision process. The first time MDP is defined is in Sec. 3, line 103.
>
> We have updated the abstract to elaborate acronyms. (Refer line 6 and line 17).
>
> > Section 1. Introduction You could define better (crisper) physical and social goals by including practical examples.
>
> The reviewer is correct. We have added an example for each in the introduction. (Refer lines 38-39 at page 1)
>
> > The authors consider social and physical goals at the same level of abstraction. However, one may figure cases where social goals drive physical goals. In these cases, social goals trigger physical goals. We can even consider, in those specific situations, that the physical goals are drives to satiate the social goals. Throughout the author's description, one can see that both concepts are disentangled, which is not (always) true - e.g. An agent wants to go towards a human to greet him (social goal); so, the agent needs to go to a certain location (physical goal) in order to greet the human. Hence, the physical goal stems from a social goal. How do you see that these cases affect your formulation? For instance, should the physical goal depend on the social goal? If so, how?
>
> We absolutely agree and believe that this framework addresses this precise issue. In the formulation we provide, social goals can depend on physical goals at different levels of recursion and vice versa. Without a doubt this is an important feature. Social goals can trigger physical subgoals and physical goals can trigger social subgoals. Indeed, in our formulation both kinds of goals equally and jointly drive the behavior of a planner, and both result in physical actions.
>
> We could make this relationship more explicit as the reviewer points out, although we note that the integration between the two is already fully-available within Social MDPs and that this would be a speedup rather than a change in the behaviour of the agents. This could be done with a hierarchical planner that proposes physical goals for both physical and social goals. We intend to do this in the future and it is likely a critical step toward a more efficient formulation.
>
> > Reward In the formula, the expresion R_j^{j-1,i} is not clear, do you mean R_j^{l-1,i}? It is the estimated reward for agent from 's point of view and following the methodology's formulation, j's are all considered with one level below the target's one.
>
> We fixed this error in equation 2. (Refer equation 2 at page 4)

---

> > ### Author Response · Authors · 2021-08-30
> > **Response to Reviewer jXz2 (Part 2 of 3)**
> >
> > > … one can envisage social behaviors included in this system, but for now it is not clear that two robots helping/hindering each other is the representation of a social interaction.
> >
> > The reviewer brings up an important theoretical issue. What is a mathematical model for social interactions? No such model exists at present, so we put forward Social MDPs for a subset of social interactions. What we mean by representation of social interactions is that the behaviors of the resulting agents are considered social behaviors by human observers. We never intend to claim that this is “the” representation of social interactions, merely that a subset of social interactions can be formulated in this manner giving rise to zero-shot execution, which is useful for both practical robotics and for hopefully making theoretical progress on this question.
> >
> >
> > > Section 2. Related Work, Simulating social interactions Rewrite lines 82-83 (word repetition - "allow").
> >
> > Fixed. (Refer line 84 to 86).
> >
> > > Section 3. Social MDPs Line 43: "Unlike I-POMDPs, Social MDPs are not recursive in terms of agent's beliefs (...)". Line 104: "Social MDPs are (...) and nested MDPs that reason about the beliefs of other agents". Please be coherent or clarify better.
> >
> > Thank you. We used beliefs in a technical manner in the first sentence and in a sloppy way in the second sentence. We have corrected this.
> > What we mean is that Social MDPs are recursive in how they estimate another agent’s reward, but not their beliefs (used in a technical sense, as in their estimates about partially-observed environments) about the world. (Refer lines 105 to 106)
> >
> > > Regarding the notion of levels, in which extent levels deeper than 2 vary from the considered ones?
> >
> > This is a good question! There are two answers here, one is trivial and the other is deeper and theoretically exciting.
> > The simple answer is that deeper levels are helpful for cases which involve more than two agents in order to chain the agent’s behaviors. Agent A wants to fix their car. Agent B wants to help Agent A. Agent C wants to help Agent B. C must estimate B’s social goals which in turn requires estimating A’s physical goal. Each additional agent in this chain requires another level to estimate another agent’s social goals. While this answer is easy, it’s not really realistic beyond a modest depth. We doubt that any human interactions require very deep chains like this one.
> > The most nuanced answer is that we did not even understand the distinction between level 1 or 2 until we began exhaustively formulating cross products of scenarios and thinking through what agents would do. Only then did we discover that level 1 and 2 lead to substantially different behavior even for pairs of agents (because you can know that another agent is trying to help you). Level 3 would allow agent X to think about how the social goals of agent Y are a consequence of X’s actions, and how the different behaviors of X can lead to changes in the social goal of Y. We are in the process of extending our work to include notions of coercion and competition which would make Level 3 much richer.
> > Theoretically, there is a practical question here. Do humans use a shortcut or do they also perform nested inference in this manner? Perhaps we can detect a signature of this with neuroimaging like fMRI, perhaps certain social disorders have to do with the level of inference being made, perhaps children develop deeper levels as they age, etc. We don’t know. But we are very excited to even be able to talk about such questions with a practical model in hand.
> >
> > > Section 3.1 Assumptions IRL should be defined and discussed. There was no previous context when "IRL" was mentioned.
> >
> > Thank you, this was a leftover from a previous model. We removed it and corrected the paragraph.
> >
> > > Section 3.2 Formal definition of Social MDPs Why the physical goal is not part of the definition? Or seen from a different perspective why the social goal is part of the definition? Being coherent here is important.
> >
> > Fixed. Refer equation 1 and lines 138 to 142.
> >
> > > Sections 3.3. ,3.4, and 3.5 Mathematical choices and formulas should be explained, discussed or referenced.
> >
> > We rewrote section 3 and went over all the equations. We added the descriptions to the equations and fixed the errors that may introduce ambiguity and inconsistency. (Refer page 4 to 5).
> >
> > > Finally, you reference Figure 3, but it is a Table. Please revise it.
> >
> > This is now Table 1. (Refer page 7)
> >
> > > Section References References 12, 19, and 30 do not contain the respective conference or journal, Please include this information.
> >
> > Fixed. (Refer references at page 9 and 10)

---

> > > ### Author Response · Authors · 2021-08-30
> > > **Response to Reviewer jXz2 (Part 3 of 3)**
> > >
> > > > Results: The results are extremely important and show how fruitful was the presented work. All images and plots must be bigger, otherwise one cannot see the real impact of your approach (the videos are also very good, but try to increase the size of the images, perhaps the inclusion of an Appendix is a great solution).
> > >
> > > We have now added the results for each of the 98 experiment scenarios in the appendix with bigger images. (Refer page 11 through 32 of appendix)
> > >
> > > > In the caption of figure 1. (b) there is a mistake: the red robot wants to hinder the yellow, not to help it.
> > >
> > > Fixed. (Refer figure 3(previously 1) at page 6)

---

### Official Review · Reviewer_BjXW · 2021-07-23

**Originality:** Very Good
**Technical Quality:** Very Good
**Clarity Of Presentation:** Very Good
**Impact:** 4

**Recommendation:**

Strong Accept: I recommend accepting the paper and will argue for my recommendation even if other reviewers hold a different opinion.

**Summary:**

This work introduces recursive social MDPs, which are able to reason about other agents physical and social goals at different levels of recursion. These social MDPs enable agents to behave more socially…for example, if another agent wants to help them, they can accept help rather than trying to do their own physical task alone. These recursive social MDPs are tested on five potential social scenarios in a gridworld, and the social MDPs predictions are shown to correlate with human predictions of social/physical goal behavior.

**Issues:**

Issues to address in the author response/revision period:
-	Why were Scenarios 1, 15, 22, 28, and 30 chosen? Are other scenarios expected to perform as well as these five?
-	How would a scenario with two helpful agents (say two agents with social goals set to 2) play out?
-	Some of the equations (for example, Equation 3), might be easier to understand with a written description alongside the mathematical equation.
-	How do the agents take joint moves in this gridworld? Does agent 1 move, and then agent 2, or do they move simultaneously?
-	The Social MDPs are tested on a task that is a fairly simple level 0 MDP. How would the time/learning complexity extend to more complex MDPs, and is it still feasible for 0 shot learning?
-	Tiny note, but I think ~R_j^(j-1) in equation 2 is supposed to read ~R_j^(l-1), is that right?


**Reviewer Expertise:**

Very good: Comprehensive knowledge of the area

**Strengths And Weaknesses:**

The proposed Social MDPs are a very interesting idea, and the stated motivation for them is strong. The proposed framework is also quite general, which lends itself well to future extensions of this work to multiple agents, and potentially POMDPs. The paper shows a good amount of thought put towards future extensions of Social MDPs.

However, it is unclear to me how the five chosen scenarios were picked for testing, and as far as I can tell there were no scenarios tested where both agents were trying to help each other. Without additional reasoning for why Scenarios 1, 15, 22, 28, and 30 were the ones chosen, it’s unclear how convincing the results are for the overall performance of Social MDPs. However, there is good performance shown on the selected five scenarios.

Overall, this is a very interesting paper, and the motivation for Social MDPs is convincing.

**Summary Of Recommendation:**

I am recommending a weak reject, because it’s unclear why the specific scenarios were picked for testing while others were excluded. However, the Social MDPs are quite interesting and well presented, and the motivation was convincing.

Updated post-rebuttal: the authors added significant experiments during the rebuttal period, including all scenarios during testing. I now recommend accepting this paper, and have changed the technical quality to Very Good. I have not seen the full written addition to the paper as the authors were not able to add the full experiments to the pdf yet, so I am somewhat torn between a weak accept and a strong accept, but I'm updating my review to a strong accept contingent on these results being added to the paper.

---

> ### Author Response · Authors · 2021-08-30
> **Response to Reviewer BjXW**
>
> We appreciate the reviewer’s thoughtful comments and have addressed these with extensive new substantive experiments.
>
> > I am recommending a weak reject, because it’s unclear why the specific scenarios were picked for testing while others were excluded. However, the Social MDPs are quite interesting and well presented, and the motivation was convincing.
>
> The reviewer is right. Rather than cherry picking we should have run every single experiment scenario. We have done so now addressing the concerns about the experiments. We have run all experiments on all scenarios, on the cross product of all of the conditions  (agents have either the same physical goal or different physical goals and one of 7 different scaling factors on each of their social goals (-2, -1, -0.5, 0, 0.5, 1, 2); 2*7*7=98 scenarios). The results for each experiment scenario can be found in the appendix. The performance for all the scenarios remained consistent with our initial results and is reported in Table 1. (Refer page 11 to 32 of appendix)
>
> > How would a scenario with two helpful agents (say two agents with social goals set to 2) play out?
>
> When both the agents have a social goal of 2 then they could either have a different physical goal (scenario 49) or a same physical goal (scenario 98). Please refer to  https://social-mdp.github.io/scenarios#scenario-49 and https://social-mdp.github.io/scenarios#scenario-98 (note: it might take longer to load the videos on the webpage). The results of these can also be found in the appendix. (Refer page 22 and 32 of the appendix).
>
> > Some of the equations (for example, Equation 3), might be easier to understand with a written description alongside the mathematical equation.
>
> The reviewer is correct, fixed. (Refer lines 138 to 190).
>
> > How do the agents take joint moves in this gridworld? Does agent 1 move, and then agent 2, or do they move simultaneously?
>
> Both agents take actions at the same time in the gridworld. When selecting the actions for agent i, we use the estimated actions for j to make decisions. The last observation is available to both agents to update the goal estimations so the action estimation is improved over time as well.
>
> > The Social MDPs are tested on a task that is a fairly simple level 0 MDP. How would the time/learning complexity extend to more complex MDPs, and is it still feasible for 0 shot learning?
>
> It is possible to run 0 shot learning on further complex MDP but the time complexity will increase as the state and action space get more complex. It is possible to speed up the model by amortized inference (similar to Netanyahu et al.’s PHASE 2021) over goals and relations by training a neural net to recognize goals and relations as initial guesses and refine them through probabilistic inference. We have updated this in the manuscript. (Refer line 201 to 203).
>
> > Tiny note, but I think ~R_j^(j-1) in equation 2 is supposed to read ~R_j^(l-1), is that right?
>
> It is. Thank you. We fixed the error. (Refer equation 2 at page 4)

---

> ### Comment · Reviewer_BjXW · 2021-09-02
> **Updated Response**
>
> The authors answered most of my questions, and I appreciated the addition of experiments on the rest of the scenarios. However, I still have concerns about the experiments. As far as I can tell, there is still only human data for the original five scenarios, and the motivation for the five scenarios chosen for collecting human data is unclear. Without extra human experiments or reason for the chosen scenarios, the human results are unclear. I have kept my review at a weak reject. Thanks to the authors for the thorough response to my questions.

---

> > ### Author Response · Authors · 2021-09-03
> > **We have collected human data and updated all of the results**
> >
> > > As far as I can tell, there is still only human data for the original five scenarios, and the motivation for the five scenarios chosen for collecting human data is unclear. Without extra human experiments or reason for the chosen scenarios, the human results are unclear.
> >
> > Ah, we misunderstood what the reviewer was asking for!
> >
> > In the past 24 hours we have collected human data for all scenarios, run our model, compared it against the human data and updated the website with the results. These will be in the final version of the manuscript (we can't update that on OpenReview right now). The full results with human data for every scenario as requested can be found here: https://social-mdp.github.io/results/
> >
> > |     Goal    | Social MDP (ours ) | Inverse Planning | Cue-based |
> > |:-----------:|:------------------:|:----------------:|-----------|
> > | Social Goal |        0.83        |       0.76       | 0.19      |
> > |   Physical Goal  |        0.74        |       0.64       | 0.06      |
> >
> > The results stay the same even though we included every single scenario thereby increased the number of scenarios several times.
> >
> > > Without extra human experiments or reason for the chosen scenarios, the human results are unclear. I have kept my review at a weak reject. Thanks to the authors for the thorough response to my questions.
> >
> > We hope that this addresses all of the reviewer's concerns.

---

> > > ### Comment · Reviewer_BjXW · 2021-09-03
> > > **Response to additional human data**
> > >
> > > Thanks for the further update, these results are much more thorough and convincing to me. This addition changes my recommendation to a weak accept, which I will update in my review.

---

> > > > ### Author Response · Authors · 2021-09-03
> > > > **Thanks!**
> > > >
> > > > Thanks for your comments!
> > > > We think that all of this resulted in a much better manuscript.

---

> ### Author Response · Authors · 2021-09-04
> **Thank You!**
>
> We thank you for your strong support in our work and help us make our manuscript stronger.
>
> In response to your post-rebuttal comments about the need for written additions to the paper, we have added all the human experiment results in the latest version of the paper here - https://social-mdp.github.io/paper/S-MDP-CoRL-Paper-Latest-Version-v1.0.pdf

---

### Official Review · Reviewer_N7Wd · 2021-07-24

**Originality:** Very Good
**Technical Quality:** Good
**Clarity Of Presentation:** Good
**Impact:** 4

**Recommendation:**

Strong Accept: I recommend accepting the paper and will argue for my recommendation even if other reviewers hold a different opinion.

**Summary:**

In response to the near complete rewrite of the methods section, and the addition of some sorely lacking figures, I have upgraded my clarity score from poor to good. After arguments with the authors, I have become convinced that in principle at least a human level of heuristic based pruning for larger numbers of agents and levels is possible in principle, although I still think there isn't a clear direction on what this heuristic might be. Due to the novelty and strong potential impact of the paper, I am upgrading my recommendation from weak accept to strong accept. The original review is left unchanged below.

===

When an has social desires i.e. desires about the success or failure of other agents, a recursive problem comes into being where each agent must estimate other agent's reward in order to estimate their own reward, which in turn requires a recursive estimation of all other agent's rewards. Traditional MDPs cannot model such agents. This paper introduces a recursive social MDP where rewards are computed at different social levels. The 0th social level excludes social rewards entirely. In the first social level, each agent is rewarded according to their 0th level, and considering the 0th level rewards of all other agents according to their social reward function. On the second level, each agent is rewarded according to the other agent's first level rewards, and so on and so forth. This paper claims to have methods for computing these social rewards, social value functions, and social policies at different levels by estimating the physical and social goals of each other agent. The paper evaluates their method by comparing its estimates of the social relationships between agents (helping or hindering) to humans estimates of the same, and finds a significantly higher correlation between these estimates and the estimate of two baseline methods.

**Issues:**

I have several significant confusions about how the method works. Many of these confusions could be resolved by the inclusion of an explicit algorithm figure for computing the actions of an agent within a social MDP and a diagram indicating how estimates of $R,Q,\tilde{\psi},\tilde{\chi},$ and $\tilde{g}$ at different levels depend on each other. It would also help a lot if each of the expressions could include a derivation, even if they are only a few lines.

After much deliberation, I now believe I understand the algorithm up to l=2 for 2 agents. I will now produce that algorithm here so that any misconceptions can be cleared up. I am deeply uncertain of many aspects of this description, especially when certain things are priors vs belief vectors at the previous level. I would like to reiterate that this was written very unclearly and in order to accept I would need a significant rewrite of the methods section.

Compute $P(\tilde{g}_2^{1,0})$ for many values of $g$, and then determine the working $\tilde{g}_2^{1,0}$ by sampling one, or using the MLE or mean or something.

=> In order to do so, estimate $\tilde{R}\_2^{0,1}$ under the assumed $g$ (which doesn't rely on $\chi$ at the 0th level), then use that to compute $Q_{2}^0$, and use that to compute $\tilde{\pi}_2^1$, the non-social policy. The non social policy can then be used to determine $P(s^{1:T} | g_2)$, and the other terms are priors at this level. This whole procedure is essentially IRL in a non-social MDP.

Under this working $g$, compute $R \rightarrow Q \rightarrow \pi$ as before. $\tilde{\psi}_2^{1,1} = \tilde{\pi}_2^1$.

Using an identical procedure, estimate $\tilde{R}_1^{2,0}$.

Compute $P(\tilde{g} _2^{1,1})$ using prior belief vector $P(\tilde{g}_2^{1,0})$ and $\tilde{\psi}_2^{1,1}$ to compute $P(s^{1:T} | g_2)$. Equation 3 does not include a level on the likelihood of $g$, so I'm assuming you pick the $\psi$ at the current level and use the $\chi$ implicitly used by $\psi$, so you use the $\chi$ of the prior level. This means that estimating $\tilde{g}_2^{1,0}$ still doesn't rely on any $\chi$ information, and so $P(\chi)$ drops away at this level as well.

Estimate $\tilde{\chi}_2^{1,1}$ using $\tilde{\psi}_2^{1,1}$ for $P(a_j^{t-1}|...)$. I assume at $t=0$ instead of recursing you use an improper prior of $1$.

Then we estimate $\tilde{R}_2^{1,1}$ using the estimated $\tilde{g}_2^{1,1}$ to determine the non-social term, $\tilde{\chi}_2^{1,1}$ as the social weight, and the earlier $\tilde{R}_1^{2,0}$ estimate of our own level 0 reward.

Finally, we can compute our own $R^2_1$ level 2 reward using our ground truth $g_1$, our ground truth $\chi_{12}$, and the estimated level 1 reward of agent 2 $\tilde{R}_2^{1,1}$. From there we can estimate our own $Q^2_1$ function, and from there our own social policy $\phi_1^2$.

I will now describe other issues with the paper.

You define $\mathcal{A} = \mathcal{A}\_i \times \mathcal{A}\_J$. Does the set J include the $i$-th agent or not? If it does, does $\mathcal{A}$ include agent $i$'s actions twice? This ambiguity is present everywhere $J$ is used, and it is used both ways e.g. in equation 4, the sum is over $j \in J, j \neq i$ which implies $i \in J$, but line 140 states "$\chi_{iJ}$ is agent $i$'s social goal towards every other agent in J." which suggests $i \notin J$.

With 2 agents, when agent 1 is trying to compute their first action, they need to maximize $Q_1^l(s, a_1, a_2, \chi_{12}) = R_1(s, a_1, a_2, \chi_{12}) + ...$, which requires knowing what $a_2$ is going to be. Agent 2 has the same problem. How is this resolved? Do the two agents take turns, and know the other agent's last action? Is there some protocol where a Pareto optimum is found?

You claim the time complexity is $O((A-1)^2 \ell)$, but you also need to solve an MDP for each value of $g$ you test at the 0th layer, which is not included in the stated complexity.

In figures 2 and 3, are the given visual bounds and +- terms a confidence interval (if so, at what confidence). If not, what do they indicate?

Line 70 "Moreover, MDPs tend to be more efficient..." What are the alternative formulations? Is this a claim you are making in this paper, or do you believe this is common knowledge? If the latter, please find a citation.

There are several notational or grammatical errors, some of which create ambiguity or affect clarity. I called out one earlier with the lack of a level on $P(g|s)$. Here are some more:

Line 91: "[24] propose" -> nit, please list the author before the citation. The sentence should still be valid if all the citations are removed.

Equation 2: $R_j^{j-1}$ -> $R_j^{l-1}

Line 197: "must compute pairwise social goal" -> nit "must compute the pairwise social goals"

Figure 1b caption "that red wants to help it" -> "that red agent wants to harm it".

Line 208: "four connected" -> omit, just include the latter description. Four connection is under specified.

**Reviewer Expertise:**

Fair: Some knowledge of the area

**Strengths And Weaknesses:**

Strengths:

The paper addresses a significant problem in using RL for human robot interactions.

The method solves a significant technical challenge.

Weaknesses:

The methods section of this paper is exceptionally unclear. I took at least 3 hours to understand just the method, in the process generating several figures that should have been in the main paper. The notation in the paper is used inconsistently, further exacerbating this problem.

Although the social MDP framework allows for arbitrary recursive levels and numbers of agents, only two agents and two levels are tested experimentally.

The method is extremely computationally expensive. $O((A-1)^2\ell)$ MDPs need to be solved, where $A$ is the number of agents and $\ell$ is the depth of the social recursion. This is impractical for all but the smallest number of agents, which is likely why the experiments are limited. The paper suggests some spatial pruning of the dependency might be possible to alleviate this, but there are many environments where our actions can affect agents spatially far away, and so it is not obvious that further speedups are possible.

**Summary Of Recommendation:**

I recommend accepting this paper contingent on a significant rewrite of the methods section, which was exceptionally unclear. This paper addresses the important problem of modelling agents who value the wellbeing of others. The method, while poorly explained, is correct to the best of my knowledge, and is a natural complete description of the problem. Social reasoning is computationally expensive and difficult for humans, and so it is not surprising that it would also be so mathematically.

I could be convinced to reject the paper if another reviewer found an unresolvable mathematical error in the method, which is reasonably likely, as I am not confident in my understanding of the method. I would also be convinced to reject if another reviewer had a strong argument that experiments with more agents or more social depth were unlikely to be as successful.

---

> ### Author Response · Authors · 2021-08-30
> **Response to Reviewer N7Wd (Part 1 of 2)**
>
> Thank you for the helpful comments, we have incorporated them into an updated manuscript and addressed them point by point below.
>
> > Many of these confusions could be resolved by the inclusion of an explicit algorithm figure for computing the actions of an agent within a social MDP
>
> We added Figure 2 to include the Social MDP algorithm sketch to show the sequence to compute the estimates and the recursion to the lower level for computing the actions. (Refer figure 2 at page 2)
>
> > A diagram indicating how estimates of R,Q,ψ~,χ~, and g~ at different levels depend on each other.
>
> We added Figure 1 to show the dependency of variables between different levels. (Refer figure 1 at page 2)
>
> > Each of the expressions could include a derivation, even if they are only a few lines.
>
> We rewrote section 3 to include the derivation of the Q function and label what each probability term means in equation 4. We also reorganized the subsections to introduce them in sequence. (Refer pages 4 through 5)
>
> > You define A=Ai×AJ. Does the set J include the i-th agent or not? If it does, does A include agent i's actions twice? This ambiguity is present everywhere J is used, and it is used both ways e.g. in equation 4, the sum is over j∈J,j≠i which implies i∈J, but line 140 states "χiJ is agent i's social goal towards every other agent in J." which suggests i∉J.
>
> The set J contains all agents in the environment so it includes i-th agent as well. We updated the definition of $\chi_{iJ}$ to make it clear that the notation is just for convenience. And we also went over all equations to make sure that we include $j \neq i$ whenever the summation is over other agents in J. (Refer line 141 to 142)
>
> > With 2 agents, when agent 1 is trying to compute their first action, they need to maximize Q1l(s,a1,a2,χ12)=R1(s,a1,a2,χ12)+..., which requires knowing what a2 is going to be. Agent 2 has the same problem. How is this resolved? Do the two agents take turns, and know the other agent's last action? Is there some protocol where a Pareto optimum is found?
>
> Both agents take actions at the same time. When selecting the actions for agent i, we use the estimated actions for j to make decisions. The last observation is available to both agents to update the goal estimations so the action estimation is improved over time as well.
>
> > The method is extremely computationally expensive. O((A−1)^2 ℓ) MDPs need to be solved, where  is the number of agents and  is the depth of the social recursion. This is impractical for all but the smallest number of agents, which is likely why the experiments are limited
>
> To address the reviewer’s concern we have run significantly more experiments referred to in the rest of the reviews. In addition, we note that quadratic algorithms are not impractical to run, moreso when the scaling factor A is small, as it is in social interactions. Most social interactions are between relatively few agents (even in a large party one does not directly and simultaneously interact with every agent). Moreover, this runtime is only required when computing the pairwise interactions between all agents, again a rare scenario at scale. Planning in general is riddled with algorithms that have relatively high complexity but are practical to run in general, e.g., AlphaZero is exponential in the branching factor of Chess, but it is still practical because not all interactions are present or meaningful.Algorithms like IPOMDPs have even far more extreme runtimes, requiring exponentially more recursive MDPs to be solved, yet they are still useful. Once the theory exists, approximate solutions can do wonders for speedups.  (Refer lines 192 to 203)
>
> > You claim the time complexity is O((A−1)^(2ℓ)), but you also need to solve an MDP for each value of g you test at the 0th layer, which is not included in the stated complexity.
>
> Absolutely. Our original formulation of the complexity was in terms of the number of recursive MDPs that must be solved, which includes the bottom MDP whose runtime scales as a function of g. We have updated the time complexity to more explicitly include the number of goals to be considered. (Refer lines 192 to 203)
>
> > In figures 2 and 3, are the given visual bounds and +- terms a confidence interval (if so, at what confidence). If not, what do they indicate?
>
> The reviewer is correct. They are 95% confidence intervals. We have updated this in the caption of the figure 4 to note this. (Refer figure 4)
>
> > Line 70 "Moreover, MDPs tend to be more efficient..." What are the alternative formulations? Is this a claim you are making in this paper, or do you believe this is common knowledge? If the latter, please find a citation.
>
> We meant this by comparison to POMDPs, we have removed this line as it was not essential.

---

> > ### Author Response · Authors · 2021-08-30
> > **Response to Reviewer N7Wd (Part 2 of 2)**
> >
> > > There are several notational or grammatical errors, some of which create ambiguity or affect clarity. I called out one earlier with the lack of a level on P(g|s). Here are some more
> >
> > We went over all the equations and fixed the errors that may introduce ambiguity and inconsistency. A level notation is included in all equations. The issues listed in the review are all fixed too.  (Refer equation 2, 3, 4, 5, 6 and 7 at page 4 and 5)
> >
> > > Line 91: "[24] propose" -> nit, please list the author before the citation. The sentence should still be valid if all the citations are removed.
> >
> > We have fixed the citation. (Refer line 91)
> >
> > > Equation 2: Rjj−1 -> $R_j^{l-1}
> >
> > Thank you, we fixed this error in equation 2. (Refer equation 2 at page 4)
> >
> > > Line 197: "must compute pairwise social goal" -> nit "must compute the pairwise social goals"
> >
> > Fixed. (Refer line 197 at page 6)
> >
> > > Figure 1b caption "that red wants to help it" -> "that red agent wants to harm it".
> >
> > Fixed. (Refer figure 3 (previously figure 1) at page 6).
> >
> > > Line 208: "four connected" -> omit, just include the latter description. Four connection is under specified.
> >
> > Updated the descriptions and included the four directions the agent can move. (Refer line 210 to 211).

---

> > ### Comment · Reviewer_N7Wd · 2021-08-30
> > **Disagreement about the number of social agents that need to be considered**
> >
> > Figures 1, 2, and the rewrite of section 3 are a significant improvement.
> > > The set J...
> > Thank you for clearing this up.
> > > we use the estimated actions for j to make decisions
> > How is the action estimated? Do you take the modal action from the estimated $\ell$-th level social policy?
> > > To address the reviewer’s concern...
> > The additional experiments are appreciated. I overstated my claim here to some extent. It's clear that humans are doing some kind of social pruning, and so in principle artificial methods must have access to some method that's at least that good.
> >
> > However, I disagree that "Most social interactions are between relatively few agents (even in a large party one does not directly and simultaneously interact with every agent)". Almost every action we take has social consequences for a large number of people. As an example, the decision to buy a certain product affects at least me, anyone consuming the product with me, the people who work at the store that sells the product, and the people who worked to produce that product. Note that spatial pruning would not do a good job of modelling interactions like these. These effects are often small or hard to estimate, and so (I assume, I am not a social scientist) humans tend to aggressively prune in the manner you've described. There are also notable circumstances where this pruning has a large cost, and it might be very valuable to have AI systems that in principle can do a better job of this than us.
> >
> > All of this is to say: I don't think the suggested pruning method is likely to work well, and it's not obvious what the better ones should be.
> >
> > > Absolutely...
> >
> > You're absorbing everything into M, yes? The change is appreciated.

---

> > > ### Author Response · Authors · 2021-08-30
> > > **The number of social agents that need to be considered**
> > >
> > > > Figures 1, 2, and the rewrite of section 3 are a significant improvement.
> > >
> > > Thanks! We agree.
> > >
> > > > However, I disagree that "Most social interactions are between relatively few agents (even in a large party one does not directly and simultaneously interact with every agent)".
> > >
> > > We think this is a very neat point that is worth exploring below. Before we start with the reviewer's example, we want to make a distinction between social interactions and interactions with social consequences.
> > >
> > > Looking down at your cellphone while you drive and hitting someone is not a social interaction. The driver did not intend to hit anyone, although they were negligent. But it has social consequences both for the driver and for the person that was hit. Just because an action has social consequences, does not make it a social interaction. Moving on to the reviewer's example:
> > >
> > > > Almost every action we take has social consequences for a large number of people. As an example, the decision to buy a certain product affects at least me, anyone consuming the product with me, the people who work at the store that sells the product, and the people who worked to produce that product.
> > >
> > > We are in complete agreement. Without a doubt this is true! But we think this proves our point. A tragic feature of how humans evaluate social interactions is that we do not consider these additional social interactions using the same mechanism that we use for small-scale social interactions. Directly forcing someone to work for you 14+ hours per day under squalid conditions without vacation for little to no pay is abhorrent and most people would consider this vile. Yet, one step removed, most people, regardless of what Amnesty International reports, don't find the same compassion and don't build the same 2nd hand social interactions with these workers. They continue to buy from such companies. The explanation for this is while an interaction exists with these workers, and it should be a social interactions, it is not, our mental mechanisms for social interactions don't apply because we don't behave as if they do.
> > >
> > > > These effects are often small or hard to estimate, and so (I assume, I am not a social scientist) humans tend to aggressively prune in the manner you've described. There are also notable circumstances where this pruning has a large cost, and it might be very valuable to have AI systems that in principle can do a better job of this than us.
> > >
> > > Perhaps what is going on is that there is a social interaction, but with some abstract concept of a "worker" and "company". These interactions seem to have far less of an emotional impact, we don't build rapport as easily, and notions of compassion don't really seem to apply in the same way. Sociologists often study face-to-face social interactions separately from other social actions.
> > >
> > > > However, I disagree that "Most social interactions are between relatively few agents (even in a large party one does not directly and simultaneously interact with every agent)".
> > >
> > > There's a question here that experiments could answer: when you have an interaction with tens of thousands of people in a supply chain, does your mental model include them? Or does it include a single averaged entity "the company". We put forward that this could be the latter. Certainly humans did not evolve with thousands of other humans in close proximity (Dunbar's number would probably be larger in that case). But this is a testable hypothesis with modern neuroimaging: larger social interactions require more compute time, we should be able to decode this from fMRI or MEG. This hasn't been done to date, but it underscores the value of a model to guide experiments.
> > >
> > > All of that being said, it could be that the reviewer is right! That humans do recruit the same mechanisms for these large-scale social interactions. In that case, we face the same problem that motion planning faces. We have a problem, we can formalize it, but it seems intractable in the real world, but humans and animals somehow solve it effortlessly. It's going to take a while to understand why this is but having the model is hand is clearly an important step.
> > >
> > > The very fact that we can have this discussion shows how understudied social interactions are and highlights dire the need for mathematical models. Our model isn't perfect by any means. It only covers some interactions. And it's computationally expensive. But it's a precisely-specified model that one can attack, disagree with, prove false, modify, that makes concrete predictions about people and what we should find when we image their brains. We think switching to this mode of research is good progress for both social robots and understanding social interactions in humans.

---

> > > > ### Comment · Reviewer_N7Wd · 2021-08-31
> > > > **Some people act on social goals wrt many people**
> > > >
> > > > So there are (at least) two concrete uses for a social MPD model: Understanding human actions, and creating a social agent. If there are humans an AI is trying to understand that take actions with regard to physically distant people, then you need to model those other people to understand the human's actions. Choosing to donate to a charity that affects people in another country is a fairly simple example of this. There are probably ways to abstract these other people into a single group object that abstracts those people, in the way that you abstract people into companies, which would reduce the effective number of people. I think there are nasty problems in e.g. coalition building/politics which cannot be easily abstracted in this way, but those may be in the minority of problems.
> > > >
> > > > On the other hand, we might imagine an agent that we want primarily to optimize for universal social goals. This agent will then have to model everyone its actions affect, and then it may have this scaling problem if even some of the people the agent interacts have many nonzero social weights. There is a sense in which this problem is limited to the difficulty of the previous, which is that (unless any human has a social goal with regard to the artificial agent) modelling the full consequences of an action on very many people only happens at level 0, and all further levels can only consider the max number of people (or abstractions of people) that actual humans can manage.
> > > >
> > > > The conceptual value of the model was never in question, my original objection was always in terms of the computational expense.

---

> > > > > ### Author Response · Authors · 2021-08-31
> > > > > **It's possible!**
> > > > >
> > > > > >  There are probably ways to abstract these other people into a single group object that abstracts those people, in the way that you abstract people into companies, which would reduce the effective number of people. I think there are nasty problems in e.g. coalition building/politics which cannot be easily abstracted in this way, but those may be in the minority of problems.
> > > > >
> > > > > Adding this level of abstraction has been one productive approach to motion planning. But for sure, there are plenty of nasty problems there. It doesn't seem like rigid abstraction layers work either here or in motion planning.
> > > > >
> > > > > > On the other hand, we might imagine an agent that we want primarily to optimize for universal social goals. This agent will then have to model everyone its actions affect, and then it may have this scaling problem if even some of the people the agent interacts have many nonzero social weights. There is a sense in which this problem is limited to the difficulty of the previous, which is that (unless any human has a social goal with regard to the artificial agent) modelling the full consequences of an action on very many people only happens at level 0, and all further levels can only consider the max number of people (or abstractions of people) that actual humans can manage.
> > > > >
> > > > > Yeah, that's very possible! The next step is to consider superhuman social interactions that involve more people and aren't limited by things like Dunbar's number. I wonder what the consequences of this would be. Maybe we can actually find cases where in small games we can exceed the social inferences that humans are willing to make.

---

> ### Author Response · Authors · 2021-09-03
> **Checking In**
>
> We were wondering if we addressed all your concerns and if you have any followup questions. Thank you!

---

> ### Author Response · Authors · 2021-09-05
> **Thank You**
>
> We thank you for your strong support of our work and helping us make our manuscript better.

---

### Author Response · Authors · 2021-09-01
**Following up with reviewers and adding the Summary of Fixes**

    Dear reviewers,

    We thank you for your valuable comments in your reviews which make a very interesting scientific discussion around the topic of social interactions. We are pleased to receive your positive enthusiasm for our work and are excited to work with you to make our manuscript a publishable paper. We are looking forward to hearing from you to see if we answered your concerns or if you have any follow-up questions. Below is the summary of all the fixes we made in the manuscript for your quick reference:


### Summary of Fixes
---
#### I. Updates to the method section (section 3):
* Included an algorithm sketch to show the sequence to compute the estimates and the recursion to the lower level for computing the actions.. (Refer figure 2 at page 2)
* Included a dependency diagram indicating the estimates of variables at different levels of reasoning. (Refer figure 1 at page 2)
* Rewrote section 3 to include the derivation of the Q function and label what each probability term means in equation 4.  (Refer page 4 and 5)
* Fixed all the equations for a few concerning notations. Also added descriptions for each equation (Refer equation 2,3,4,5,6,7 in page 4 and 5)
* Reorganized the subsections of section 3 to introduce them in sequence. (line 138 to 190).

#### II. Experiments
We ran 98 new experiments on all scenarios, on the cross product of all of the conditions  (agents have either the same physical goal or different physical goals and one of 7 different scaling factors on each of their social goals (-2, -1, -0.5, 0, 0.5, 1, 2); 2*7*7=98 scenarios). The results for each experiment scenario can be found in the appendix. The performance for all the scenarios remained consistent with our initial results and is reported in Table 1.  (Refer Page 11 through 32 of the appendix).

#### III. Fixed other fixes
Addressed 32 other issues raised by the reviewers and have fixed them in the manuscript. The line/page numbers are added to each review response to the reviewer.

---

> ### Author Response · Authors · 2021-09-03
> **Addressed further comments**
>
> In response to comments in the past day we have run an extensive human evaluation and made formatting changes.
> These were the last two comments of two reviewers. We believe that we've addressed every comment with substantive changes.

---

### Meta-Review · Area_Chair_9L89 · 2021-08-14

**Recommendation:** Accept (Poster)
**Confidence:** 4

**Metareview:**

### Final Meta-Review

The authors have thoughtfully and proactively engaged with the concerns raised by reviewers and have made significant improvements to the paper, including substantially clearer technical exposition and a rich set of new simulation results to more rigorously evaluate the proposed framework.

Ultimately, the final manuscript has earned strong reviewer support and will no doubt be a welcome contribution to CoRL. I am pleased to recommend the paper for acceptance.

### Original Meta-Review

The reviewers agree that the paper addresses a significant and well-motivated problem and proposes a substantially novel approach to tackle it. On the other hand, they also raise concerns regarding clarity of exposition and adequacy of the experimental evaluation.

All three reviewers have stressed the lack of clarity in the Results section, as well as inconsistencies or typos in the mathematical notation. This section would need to be significantly revised by the authors prior to eventual publication of the paper. Reviewers have also suggested the use of examples and visual illustrations to improve the paper's readability.

Reviewer N7Wd brings up the challenging scalability of the proposed formulation with the number of agents and levels considered and notes that only two agents and two levels (the bare minimum) are shown. Reviewer BjXW notes the lack of justification regarding the five scenarios chosen, as well as the apparent absence of a mutually helpful scenario.

Finally, I strongly encourage the authors to clearly establish the connection between their proposed formulation and dynamic (Markov) games, especially game-theoretic formulations that model "altruism" and "spite" as part of players' payoffs.

---

> ### Author Response · Authors · 2021-08-30
> **Response to Meta Review (Part 1 of 2)**
>
> > The reviewers agree that the paper addresses a significant and well-motivated problem and proposes a substantially novel approach to tackle it. On the other hand, they also raise concerns regarding clarity of exposition and adequacy of the experimental evaluation.
>
> We appreciate the note. We have expanded the evaluation, making it 10 times larger, and performing every cross product experiment that the reviewers asked for. We also addressed all of the exposition notes with an updated manuscript.
>
> >All three reviewers have stressed the lack of clarity in the Results section, as well as inconsistencies or typos in the mathematical notation. This section would need to be significantly revised by the authors prior to eventual publication of the paper. Reviewers have also suggested the use of examples and visual illustrations to improve the paper's readability.
>
> We appreciate the notes and have resolved clarity issues, inconsistencies and typos. We have significantly revised the Results section and added additional examples and illustrations.
>
> > Reviewer N7Wd brings up the challenging scalability of the proposed formulation with the number of agents and levels considered and notes that only two agents
>
> We did focus on social interactions between dyads, pairs of agents, but we note that most social interactions are between pairs and an understanding of pairwise interactions is easily extended to more agents (of course with associated computational costs). Most algorithms in planning have difficulty scaling to many agents without some approximation method, we think having the pairwise formulation is the first step to discovering algorithms that are able to scale well.
>
> > two levels (the bare minimum) are shown.
>
> We should have been clearer. We present three levels (level 0, level 1, and level 2). Roughly:
> Level 0 “I want to move this block” Level 0 corresponds to an MDP, this is a baseline.
> Level 1 “I want to help this person who is moving the block” is already social, it corresponds to agents that are able to reason about each other’s physical goals. This allows agents to help or hinder one another, but they treat one another as if the other agent doesn’t have their own social goals.
> Level 2 “I want to stop this person from being helpful” is an even deeper social level where agents are able to reason about the social goals of other agents.
> This goes well beyond the minimum number (which is two) and demonstrates the utility of our method, deeper nesting levels correspond to more sophisticated social interactions.
>
> > Reviewer BjXW notes the lack of justification regarding the five scenarios chosen, as well as the apparent absence of a mutually helpful scenario.
>
> We address this by running experiments for every single cross product of physical goal and social goal configuration which leads to 98 possible scenarios. Each scenario has agents as having either the same physical goal or different physical goals and one of 7 different scaling factors on each of their social goals (-2, -1, -0.5, 0, 0.5, 1, 2) which is (2*7*7=98 scenarios). The results of each experiment scenario can be found in the appendix along with the results reported in Table 1.
> A mutually helpful scenario is possible in two ways.When both the agents have a social goal of 2 then they could either have the same physical goal (scenario 49) or a different physical goal (scenario 98). The results of these can also be found in the appendix (page 11 through 32 of the appendix). Also, the interactions for these two scenarios are https://social-mdp.github.io/scenarios#scenario-49 and https://social-mdp.github.io/scenarios#scenario-98.

---

> > ### Author Response · Authors · 2021-08-30
> > **Response to Meta Review (Part 2 of 2)**
> >
> > > Finally, I strongly encourage the authors to clearly establish the connection between their proposed formulation and dynamic (Markov) games, especially game-theoretic formulations that model "altruism" and "spite" as part of players' payoffs.
> >
> > This is a great point! We should have done that initially. The two are related, just like a game-theoretic analysis of chess is related to a planner which actually plays chess. In game theory one specifies the utility/payoff of the agents playing a game. Following the popular approach of Levine (1998), to encode altruism or spite one can mix the payoffs for multiple agents in the utility for each agent. One obtains altruism/prosocial behavior when using positive coefficients for the other agent’s payoff and spitefulness/antisocial behavior when using negative coefficient for the other agent’s payoff. This is closely related to our approach, where we perform similar mixing of the agent’s reward functions. In addition, game theory and planning are closely related, one can often translate goals and behaviors from the mathematical language of one to the other.
> >
> > With this in mind there are two key differences between our approach and that taken in game theory.
> >
> > The first is theoretical. Social MDPs generalize the notion of altruism and spitefulness commonly used in game theory. Formulating the altruism or spitefulness of agent X by including the payoff of agent Y into X’s own payoff corresponds to a level 1 model in Social MDPs. X cares about Y’s direct payoff, but not about Y’s own altruism or spitefulness toward other agents. Practically, a level 2 model would allow X to help Y help other agents. This could be translated back into a game-theoretic model that would extend the work of Levine 1998 to a richer set of scenarios and behaviors.
> >
> > The second is practical. Models of social behavior in game theory specify the payoffs of other agents directly. In other words, to formulate the payoff of an altruistic agent X you create a linear combination of the total earnings of agent X and agent Y with the magnitude of the (positive) coefficient in the linear combination determining how altruistic X is. In our formulation, one includes the reward of another agent, rather than the precise payoff.  This second point is what enables Social MDPs to drive social robots in diverse environments where agents can have any combination of goals. The game-theoretic formulation requires that the goals be known ahead of time in order to set the payoffs, severely limiting practical applications, since robots won’t know the other agent’s goals ahead of time.
> >
> > We have included a note about game theory in the related work section. (Refer lines 72 to 75)

---

> > > ### Comment · Area_Chair_9L89 · 2021-09-09
> > > **A Level 2 response (or Meta-Response to the Response to the Meta-Review)**
> > >
> > > Thank you for the thoughtful discussion on this point, and for the condensed version now included in the updated manuscript.
> > >
> > > That said, I think your statement about the assumption of mutual payoff knowledge made by game theory is inaccurate: games with private player payoffs have been extensively considered and studied in the field (e.g. auction games, Bayesian games...). Assistance games (a.k.a. cooperative inverse reinforcement learning [13]) are a recent special case that you are surely familiar with: in these (Markov) games, the "robot" is a fully altruistic player who is, however, uncertain about the payoff of the "human" player whom it seeks to help. No assumption of a-priori-known goals is made here, the robot seeks to learn the human's payoff over time and the human is in turn incentivized to teach it.
> > >
> > > In fact, your statement about game theory being concerned with Level 1 only is also inaccurate as far as I understand it. The definition of a CIRL game equilibrium in [13] can be seen as the limit as k goes to infinity of the Level k agent policies in a Social MDP, since it is the pair of strategies played when the human wants to succeed at their objective, the robot wants to help the human, the human wants to help the robot help the human, and so on ad infinitum.
> > >
> > > The above is also called a pragmatic-pedagogic equilibrium, precisely because the human is trying to pedagogically help the robot learn how to help them, and the robot is in turn pragmatically interpreting the human's actions as coming from such a pedagogically-minded agent (i.e. the robot wants to help the human teach the robot how to help the human, which is already establishing a Level 3 type objective).
> > >
> > > In short, I think the connection between Social MDPs and dynamic game theory is deeper than you may be giving it credit for.
> > >
> > >
> > > Finally, I caught a couple of small typos.
> > >
> > > * Line 74: a prior → a priori
> > > * Line 105: of other agent's → of other agents'

---

> ### Author Response · Authors · 2021-08-30
> **Summary of Fixes**
>
> ## Summary of Fixes
> ---
> ### I. Updates to the method section (section 3)
> * Included an algorithm sketch to show the sequence to compute the estimates and the recursion to the lower level for computing the actions.(Refer figure 2 at page 2)
> * Included a dependency diagram indicating the estimates of variables at different levels of reasoning. (Refer figure 1 at page 2)
> * Rewrote section 3 to include the derivation of the Q function and label what each probability term means in equation 4.  (Refer page 4 and 5)
> * Fixed all the equations for a few concerning notations. Also added descriptions for each equation (Refer equation 2,3,4,5,6,7 in page 4 and 5)
> * Reorganized the subsections of section 3 to introduce them in sequence. (line 138 to 190).
>
> ### II. Experiments
> Ran 98 new experiments on all scenarios, on the cross product of all of the conditions  (agents have either the same physical goal or different physical goals and one of 7 different scaling factors on each of their social goals (-2, -1, -0.5, 0, 0.5, 1, 2); 2\*7\*7=98 scenarios). The results for each experiment scenario can be found in the appendix. The performance for all the scenarios remained consistent with our initial results and is reported in Table 1.  (Refer Page 11 through 32 of the appendix).
>
> ### III. Fixed 32 other issues
> We addressed 32 other concerns raised by the reviewers and updated the manuscript accordingly. The line/page numbers added to each review response to the reviewer.

---

### Decision · Program_Chairs · 2021-09-13

**Decision:**

Accept (Poster)

**Comment:**

### Final Meta-Review

The authors have thoughtfully and proactively engaged with the concerns raised by reviewers and have made significant improvements to the paper, including substantially clearer technical exposition and a rich set of new simulation results to more rigorously evaluate the proposed framework.

Ultimately, the final manuscript has earned strong reviewer support and will no doubt be a welcome contribution to CoRL. I am pleased to recommend the paper for acceptance.

### Original Meta-Review

The reviewers agree that the paper addresses a significant and well-motivated problem and proposes a substantially novel approach to tackle it. On the other hand, they also raise concerns regarding clarity of exposition and adequacy of the experimental evaluation.

All three reviewers have stressed the lack of clarity in the Results section, as well as inconsistencies or typos in the mathematical notation. This section would need to be significantly revised by the authors prior to eventual publication of the paper. Reviewers have also suggested the use of examples and visual illustrations to improve the paper's readability.

Reviewer N7Wd brings up the challenging scalability of the proposed formulation with the number of agents and levels considered and notes that only two agents and two levels (the bare minimum) are shown. Reviewer BjXW notes the lack of justification regarding the five scenarios chosen, as well as the apparent absence of a mutually helpful scenario.

Finally, I strongly encourage the authors to clearly establish the connection between their proposed formulation and dynamic (Markov) games, especially game-theoretic formulations that model "altruism" and "spite" as part of players' payoffs.